# Quantum-mechanical exploration of the phase diagram of water

Aleks Reinhardt [1✉] & Bingqing Cheng [2,3✉]

The set of known stable phases of water may not be complete, and some of the phase boundaries between them are fuzzy. Starting from liquid water and a comprehensive set of 50 ice structures, we compute the phase diagram at three hybrid density-functional-theory levels of approximation, accounting for thermal and nuclear fluctuations as well as proton disorder. Such calculations are only made tractable because we combine machine-learning methods and advanced free-energy techniques. The computed phase diagram is in qualitative agreement with experiment, particularly at pressures ≲ 8000 bar, and the discrepancy in chemical potential is comparable with the subtle uncertainties introduced by proton disorder and the spread between the three hybrid functionals. None of the hypothetical ice phases considered is thermodynamically stable in our calculations, suggesting the completeness of the experimental water phase diagram in the region considered. Our work demonstrates the feasibility of predicting the phase diagram of a polymorphic system from first principles and provides a thermodynamic way of testing the limits of quantum-mechanical calculations.

[1] Department of Chemistry, University of Cambridge, Lensfield Road, Cambridge CB2 1EW, UK. [2] Accelerate Programme for Scientific Discovery, Department of Computer Science and Technology, 15 J.J. Thomson Avenue, Cambridge CB3 0FD, UK. [3] Cavendish Laboratory, University of Cambridge, J.J. Thomson Avenue, Cambridge CB3 0HE, UK. ✉email: ar732@cam.ac.uk; bc509@cam.ac.uk

Water is the only common substance that appears in all three states of aggregation—gas, liquid and solid—under everyday conditions[1], and its polymorphism is particularly complex. In addition to hexagonal ice (ice $I_h$) that forms snowflakes, there are currently 17 experimentally confirmed ice polymorphs and several further phases have been predicted theoretically[2]. The phase diagram of water has been extensively studied both experimentally and theoretically over the last century; nevertheless, it is not certain if all the thermodynamically stable phases have been found, and the coexistence curves between some of the phases are not well characterised[2]. At high pressures, experiments become progressively more difficult, and therefore computer simulations play an increasingly crucial role.

Computing the thermodynamic stabilities of the different phases of water is challenging because quantum thermal fluctuations and, in proton-disordered ice phases, the configurational entropy need to be taken into account in free-energy calculations. Considerable insight has been gained into the phase behaviour of water using empirical potentials[3–13], which inevitably entail severe approximations[3]. For example, the rigid water models such as TIP$n$P[14–16] and SPC/E[17] cannot describe the fluctuations of the bond lengths and angles and therefore do not explicitly include nuclear quantum effects (NQEs)[18,19], although some have been extended to incorporate fluctuations[20–22]. The MB-pol force field[23], which includes many-body terms fitted to the coupled-cluster level of theory, has not been fitted to the high-pressure part of the water phase diagram. Describing the phase diagram is a particularly stringent test for water models, and indeed, only the TIP4P-type models[14–16] and the iAMOEBA water model[24] reproduce the qualitative picture, while the SPC/E, TIP3P and TIP5P models predict ice $I_h$ to be stable only at negative pressures[25,26].

A promising route for predicting phase diagrams is from electronic structure methods (i.e. ab initio), but combining these methods with free-energy calculations is extremely expensive. However, machine-learning potentials (MLPs) have emerged as a way of sidestepping the quantum-mechanical calculations by using only a small number of reference evaluations to generate a data-driven model of atomic interactions[27]. As an example, MLPs have been employed to reveal the influence of Van der Waals corrections on the thermodynamic properties of liquid water[28]. Later, a similar framework, also employing accurate reference data at the level of hybrid density-functional theory (DFT), reproduced several thermodynamic properties of solid and liquid water at ambient pressure[29]. Very recently, multithermal–multibaric simulations were used to compute the phase diagram and nucleation behaviour of gallium[30], while simulations using MLPs provided evidence for the supercritical behaviour of high-pressure hydrogen[31].

Before performing free-energy calculations to compute the phase diagram, one must decide which ice phases to consider, bearing in mind that the experimentally confirmed phases may not be exhaustive. Moreover, many ice phases come in pairs of a proton-disordered and proton-ordered form, e.g. $I_h$ and XI, III and IX, V and XIII, VI and XV, VII and VIII, and XII and XIV[32–34]. Ices III and V are known to exhibit only partial proton disorder[35]. Both full and partial proton disorder make free-energy calculations more challenging. This is because the enthalpies of different proton-disordered manifestations can be significantly different, but the disorder is not possible to equilibrate at the time scale of molecular dynamics (MD) simulations.

Here, we compute the phase diagram of water at three hybrid DFT levels of theory (revPBE0-D3, PBE0-D3 and B3LYP-D3), accounting for thermal and nuclear fluctuations as well as proton disorder. We start from 50 putative ice crystal structures,

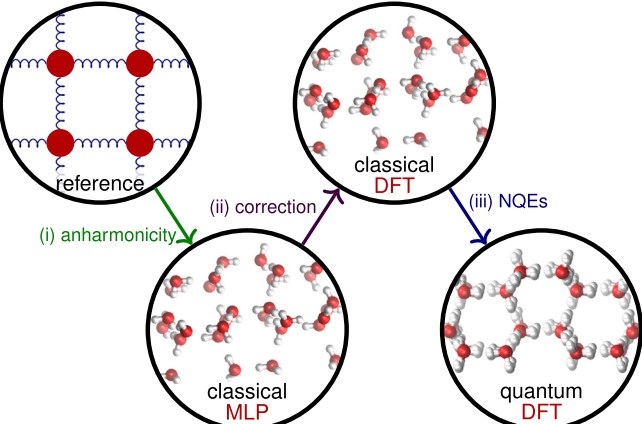

**Fig. 1 Schematic illustration of the procedure.** Thermodynamic integration steps from a reference system to the classical system described by the MLP, to the classical system described by DFT, and finally to the system with quantum-mechanical nuclei described by DFT.

including all the experimentally known ices. To circumvent the prohibitive cost of ab initio MD simulations, we use a recent MLP[29] as a surrogate model when performing free-energy calculations, and then promote the results to the DFT level as well as account for NQEs. This workflow is illustrated in Fig. 1 and described in the 'Methods' section.

## Results

**Chemical potentials.** For each ice structure, we first obtain the chemical potential over a temperature range of 25 K to 300 K and a pressure range of 0 bar to 10,000 bar by performing classical free-energy calculations using the thermodynamic integration (TI) method as described in the 'Methods' section. The MLP employed in these calculations is based on the hybrid revPBE0[36] functional with a semi-classical D3 dispersion correction[37]. This MLP reproduces many properties of water, including the densities and the relative stabilities of $I_h$, $I_c$ and liquid water at the ambient pressure[29]. Because the MLP is only trained on liquid water and not on either the structures or the energetics of the ice phases, it allows us to explore the ice phase diagram in an agnostic fashion. It nevertheless reproduces lattice energies, molar volumes and phonon densities of states of diverse ice phases, since local atomic environments found in liquid water also cover those observed in the ice phases[38].

The chemical potentials (expressed per molecule of $H_2O$ throughout this work) at 225 K computed using the MLP are shown in Fig. 2a. Focussing only on the phases whose MLP chemical potential is within 10 meV of the ground-state phase under all conditions of interest, we narrow down the selection to 12 ice phases—ices $I_h$, $I_c$, II, III, V, VI, VII, IX, XI, $XI_c$, XIII and XV—and the liquid. These phases are all known from experiment, which suggests that the experimental phase diagram of water is indeed complete[2]. Moreover, as the chemical potential difference between $I_h$ and $I_c$ is small and has already been studied in ref. [29], we report the results only for 'ice I'.

The treatment of proton-disordered phases is complicated because choosing a non-representative configuration can introduce a significant bias in enthalpy and in turn lead to an incorrect phase diagram[8]. To overcome this, we use a combination of the Buch algorithm[39] and GenIce[40] to generate 5–8 different depolarised proton-disordered manifestations of each disordered phase considered. For the partially proton-disordered phases III and V, we first construct many configurations, and then only consider the ones that match the experimental site occupancies[41].

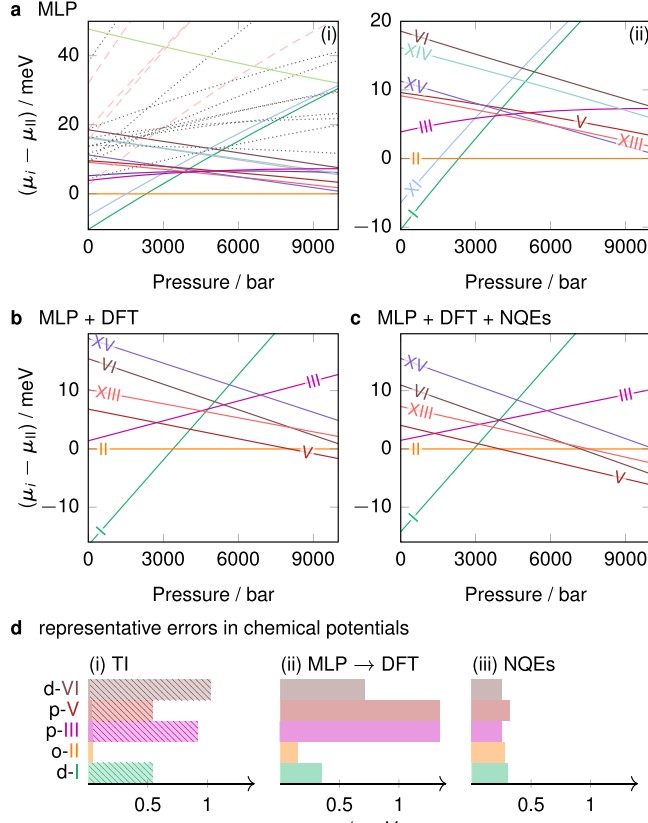

**Fig. 2 Computed chemical potentials and error analysis.** Per-molecule chemical potentials for the phases (meta)stable at 225 K, shown relative to the chemical potential of ice II. The MLP results are in **a**. In subpanel (i), solid lines show experimentally known bulk phases, dashed lines show various known low-density structures, and dotted lines show other hypothetical phases. Subpanel (ii) shows the same data, but only for the known experimental phases, as labelled. **b** Shows the chemical potentials at the level of the revPBE0-D3 DFT for classical systems, and **c** shows DFT results with NQE corrections. **d** Illustrates representative standard deviations for selected proton-disordered ('d'), partially proton-disordered ('p') and proton-ordered ('o') phases. Contributions are split into those arising from (i) the thermodynamic integration to the classical system described by the MLP, with the contribution from proton disorder shown as a lighter hatched area; (ii) the MLP to the revPBE0-D3 correction term; and (iii) the term accounting for NQEs.

We compute the chemical potential of each such configuration independently and average over the results, and finally add a configurational entropy associated with (partial) proton disorder[34,35]. We have assumed that the experimental proton disorder is correct for the potential used, which may not be completely accurate[8,42], but is the best assumption we can make, since equilibration of proton disorder is not feasible to achieve on computationally tractable time scales.

We perform a free-energy perturbation to promote the MLP results to the hybrid DFT level (see revPBE0-D3 results in Fig. 2b), as detailed in the 'Methods' section. This correction is necessary to recover the true chemical potential at the DFT level, because the MLP inevitably leads to small residual errors[43]. Specifically, proton order leads to long-range electrostatics that can destabilise the solid, but the MLP only accounts for short-range interactions. Albeit small in absolute terms, the correction to DFT causes a changeover of stability, and in particular reduces the stability of the proton-ordered phases (such as ice II and XV), as can be seen from Fig. 2b. Although the proton-ordered phases

XI, XIII and XV are more stable than their proton-disordered analogues (I, V and VI, respectively) at the MLP level at low temperatures, at the DFT level, the proton order–disorder transition occurs at considerably lower temperatures than we focus on here, and so we do not characterise this transition further, and we do not show the proton-ordered phases XI or XIV in Fig. 2b.

Finally, we consider NQEs by performing path-integral molecular dynamics (PIMD) simulations. In general, NQEs serve to stabilise higher-density phases. For all phases, the degree of stabilisation increases with increasing pressure, and decreases with increasing temperature. Whilst the overall form of the chemical potential plot (Fig. 2c) changes only subtly, NQEs can significantly shift the phase boundaries, and we return to this point below.

Although it is often difficult to determine error bars in free-energy calculations[44], we have estimated the errors arising from each step of the calculation (Fig. 2d) following the steps outlined in the 'Methods' section. The uncertainty is small for the proton-ordered phases, but considerably larger for the proton-disordered phases, in which there are more varied local environments. The typical uncertainty in the chemical potential of each phase is at most ~2.5 meV, but, as we show below, even this relatively small difference is often sufficient to change the coexistence lines significantly.

**Phase diagram.** We determine the phase diagram of water by analysing the computed chemical potentials over a wide range of pressure and temperature conditions. In Fig. 3, we show the phase diagram for the classical system described by the MLP, the ab initio phase diagram including NQEs, as well as the experimental phase diagram for comparison. Even just at the MLP level without NQEs, the phase diagram is already a reasonable approximation (Fig. 3b). It captures the negative gradient of the pressure–temperature coexistence curve between ice I and the liquid, but fails to account for ices III and V, and the proton-ordered ice XV is more stable than its disordered analogue (VI). As discussed above, the MLP is prone to overestimating the stability of the proton-ordered phases, which leads to this computational artefact.

When the MLP is corrected to the DFT level of theory and NQEs are accounted for, the resulting phase diagram (Fig. 3c) is considerably improved, and is in close agreement with experiment. [The computed chemical potential difference between ice I and water at the same level of theory in ref. [29] suffered from a flipped sign when adding the $\mu^{MLP \to DFT}$ terms, and the corrected melting point for $H_2O$ is 274(2) K. The error bar on the melting point of the current work is larger at ~6 K, primarily because only 5 different realisations of the proton-disordered ice $I_h$ were considered rather than the 16 of ref. [29].] However, at very high pressure, the coexistence curve between ices V and VI has too steep a gradient compared to experiment. This may arise from the inaccuracy of the reference DFT functionals[45] or from the finite basis set and energy cutoffs employed in the DFT calculations. To illustrate the effect of employing different DFT approximations, we show analogous phase diagrams for two alternative DFT functionals, B3LYP-D3 and PBE0-D3, in Fig. 3d. The melting points at 1 bar, 274(6) K and 268(6) K, respectively, are somewhat lower than for the revPBE0-D3 functional, although the error bars are comparable with the difference. In terms of the phase diagrams at higher pressures, PBE0-D3 shows better agreement with experiment compared to the B3LYP-D3 functional.

In none of three phase diagrams shown in Fig. 3c, d is ice III thermodynamically stable. From the chemical potential results, we can see that ice III is within ~2 meV to the thermodynamically stable phase at pressures around 3000 bar and temperatures

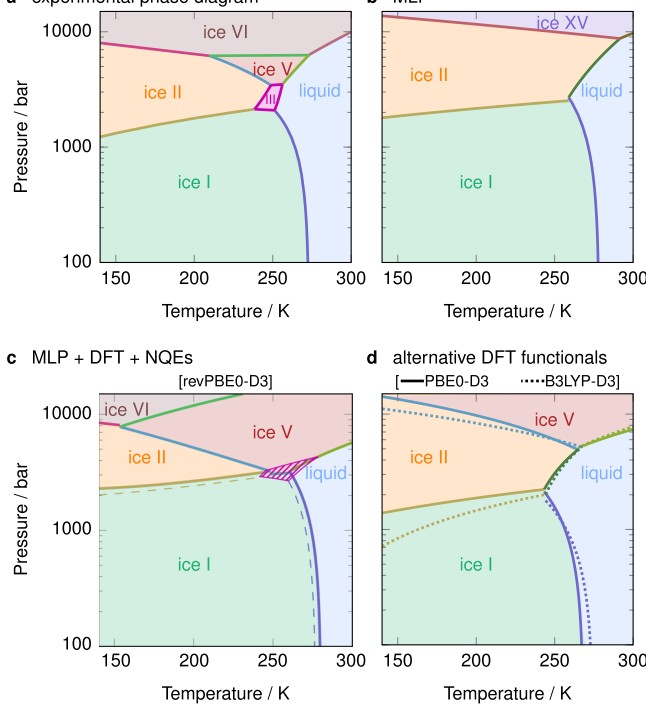

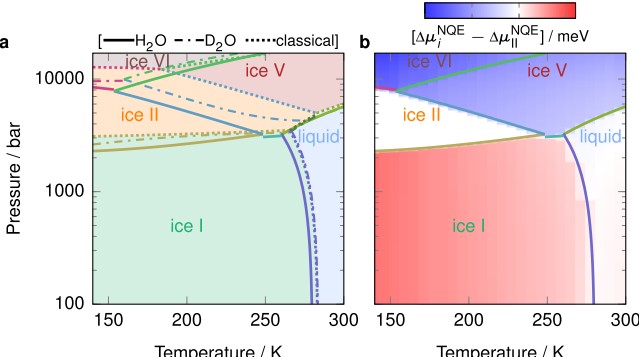

**Fig. 4 Phase diagram dependence on NQEs. a** Phase diagram excluding NQEs (dotted lines) alongside the full phase diagram (solid lines from Fig. 3c) and the phase diagram for $D_2O$ (dotted–dashed lines). **b** Heat map indicating the correction to the chemical potential of the stable phase $i$ due to NQEs, relative to ice II, overlaid on the phase diagram from Fig. 3c.

**Fig. 3 Comparison of phase diagrams.** Phase diagrams **a** from experiment[64, 65] and **b**–**d** from simulation. The phase diagram in **b** corresponds to the classical system described by the MLP, while **c**, **d** are at the indicated DFT levels and also account for NQE corrections. Dashed lines in **c** correspond to an alternative calculation where the chemical potential of ice III was decreased and that of ice I was increased by the typical error in chemical potential calculations (Fig. 2d), as discussed in the text. The hatched area corresponds to the region of stability of ice III in this alternative calculation.

around 250 K, where it is known to be stable from experiment. However, its chemical potential depends crucially on the particular manifestation of proton disorder that we choose, and to compute the chemical potentials of the proton-disordered ice phases, we average over a number of independent simulations with different initial proton-disorder configurations. As we have shown in Fig. 2d, the typical error in computations associated with proton-disordered phases is about 2 meV. The extent to which such uncertainties affect the computed phase diagram can be significant: we show in Fig. 3c the phase diagram obtained when ice III is stabilised whilst ice I is destabilised, in each case by changing their chemical potentials by the standard deviation obtained from the above procedure. In such a scenario, ice III is stable over approximately the right range of temperatures and pressures. Such small changes in free energy particularly affect partially proton-disordered phases such as ice III: different partial orderings (i.e. those with different proton site occupancies) may be stable to different extents[8,42], and the calculated stability thus depends on the specific proton site occupancies considered in addition to the uncertainty in the calculation itself. The main lessons to be drawn from this are that (i) many of the phases are very close in free energy, particularly in the region around 250 K and ~3000 bar, which means that even if they are not thermodynamically stable, they are likely to be metastable and hence may be easier to obtain experimentally at these conditions, and (ii) the phase diagram is an extremely sensitive test of the underlying potential: even a very small change in relative stability can drastically affect the phase behaviour.

Finally, we show the phase diagram with and without accounting for NQEs in Fig. 4a. The NQE correction to the chemical potential of the most stable phase is shown in Fig. 4b; we show the results relative to (metastable) ice II at the same pressure and temperature in order to remove the large effect of the dependence of the mean kinetic energy on the temperature. As we have already discussed, the addition of NQEs to the calculation stabilises the denser phases relative to the less dense ones: ice VI is more stabilised than ice V, ice V more than ice II, ice II more than the liquid, and the liquid more than ice I. We also note that for the solid phases, the density of each phase correlates well with the tetrahedrality parameter studied in this context in ref. 22.

When NQEs are not accounted for, $H_2O$ and $D_2O$ exhibit exactly the same thermodynamic behaviour. However, accounting for the difference between them is necessary for example, for polar ice dating and historical atmospheric temperature estimation[46], which suggests that a classical description of nuclear motion is not satisfactory. We thus account for NQEs by integrating Eq. (2) to the mass of hydrogen or deuterium to determine the phase behaviour of light and heavy water, respectively. We show the predicted phase diagram of $H_2O$, $D_2O$ and 'classical' water (without NQEs) in Fig. 4a. The melting point of ice I for $H_2O$ is approximately 4 K lower than that for $D_2O$, as investigated previously[29,47]. It is interesting that $D_2O$ follows almost the same ice I–liquid coexistence line as classical water, since positive and negative contributions to the integral of Eq. (2) largely cancel out[29]. A similar conclusion can be drawn about the coexistence line between ices I and II, perhaps because the density difference between the two phases is similar to that between ice I and the liquid. At higher pressures, however, the phase diagrams of $H_2O$ and $D_2O$ are drastically different from classical water, demonstrating the importance of properly accounting for NQEs.

## Discussion
In this work, we have undertaken an exhaustive exploration of the phase diagram of water at three levels of hybrid DFT, including nuclear thermal fluctuations and proton disorder. The agreement between the computed and experimental phase diagrams is very good, but by no means perfect. Part of the difference can be explained in terms of the sensitivity to very small changes in the chemical potentials, which is particularly acute with proton-disordered phases. Furthermore, using DFT to model water, even at the hybrid level, has limitations[45]. Finally, it is not guaranteed

that the phases obtained in experiment are in fact the thermo-dynamically stable phases; as such, the experimental and the theoretical phase diagram may have subtle differences arising from slightly different definitions of phase stability. Nevertheless, the ab initio phase diagrams show a significant improvement over previous results based on empirical models[24,26], particularly at low pressure. Moreover, although NQEs are often neglected in the computation of the phase diagram of water, we have properly accounted for them in the present work. This allows us to characterise further the subtle difference between the phase behaviour of $H_2O$ and $D_2O$.

Our entire calculation was premised on the fact that a MLP had been parameterised for water[29], which permitted the study of larger systems for sufficiently long to be able to compute their thermodynamic properties. While the DFT-level phase behaviour we have studied is the best we can do at this stage, for many applications, such as investigating the thermodynamics of the nucleation of ice, even the approach we have followed may not be computationally feasible. It would therefore be very tempting to use the inexpensive MLP on its own in future studies. From our results, we can conclude that, even though the potential was trained solely on liquid-phase structures, the description of the solid phases and the phase behaviour of the MLP on its own is very reasonable. However, a degree of care must be taken in interpreting its results, not least because proton-ordered phases are somewhat too stable when described by the MLP.

The good agreement between the calculated phase diagram and experiment confirms that the hybrid DFT levels of theory describe water well. In fact, the approach we have outlined to compute free energies and in turn phase diagrams provides a particularly difficult benchmark for quantum-mechanical methods. We have shown that three different hybrid DFT functionals (revPBE0-D3, PBE0-D3, B3LYP-D3) result in similar, but certainly not identical, phase behaviour. It would be interesting to apply the same workflow to other electronic structure methods, including random-phase approximation[48] and DFT at the double hybrid level[49]. Indeed, in the future, one possible way of bench-marking and optimising DFT functionals may well be to evaluate the phase diagram of the material studied.

The present study bridges the gap between electronic structure theory and accurate computation of phase behaviour for water. With a robust framework in place, we aim in future work to investigate the behaviour of ice that is less well understood experimentally. For example, tantalisingly, in Fig. 2a, the chemical potentials of the low-density phases of ice have a large gradient; it would be interesting to determine the phase diagram at negative pressures, which remains rather less well explored[50]. Another intriguing phase transition to study is that between ice VII and liquid water at higher pressures than we have looked at here, which empirical pair potentials do an especially poor job of describing[4,51]. Moreover, the same framework can be used to investigate (meta)stability of novel bulk materials. It further lays the groundwork for utilising experimental phase diagrams to correct and improve interatomic potentials based on electronic structure methods.

## Methods
**Candidate ice structures**. We start from the 57 phases that were screened from an extensive set of 15,859 hypothetical ice structures using a generalised convex hull construction, an algorithm for identifying promising experimental candidates[52,53]. We eliminate defected phases, dynamically unstable phases, and the very high pressure phase X, but add the originally missing ice IV. In Supplementary Data 1, we provide the structures of the remaining 50 ice phases that we consider in our simulations.

**DFT calculations**. We compute the energies of the water structures using the CP2K code[54] with the revPBE0-D3, the PBE0-D3 and the B3LYP-D3 functionals.

The computational details of the calculations are identical to refs. [29,55] but for the choice of functionals, and we provide input files in Supplementary Data 1.

**Free-energy calculations**. We use the method of thermodynamic integration (TI) to compute the Gibbs energy of the selected ice phases at a wide range of ther-modynamic conditions, which in turn determine the ice phase diagram. A similar method has previously been used to compute the ice phase diagram using empirical potentials[4]. In our approach, we use a series of steps in simulations of physical or artificial systems to compute the various components of the free-energy difference between a reference system and the fully anharmonic, quantum system, as depicted schematically in Fig. 1.

In the first step of the free-energy calculation for the solid phases, we equilibrate a sample of an ice phase at the temperature and pressure of interest in an isothermal-isobaric simulation with a fluctuating box with a Parrinello–Rahman-like barostat[56] to determine the equilibrium lattice parameters at the conditions of interest. We then prepare a perfect crystal with the corresponding lattice parameters and minimise its energy, compute the Helmholtz energy of the reference harmonic crystal by determining the eigenvalues of the hessian matrix, account for the motion of the centre of mass, and finally perform a thermodynamic integration step to the potential of interest in which classical nuclei are described by the MLP [Fig. 1(i)]. By adding a suitable pressure–volume term, we obtain the fully anharmonic classical chemical potential of the ice system described by the MLP. The details of this procedure are discussed in ref. [57]. In this step, we use reasonably large system sizes (of the order a few hundred to several thousand water molecules) to ensure that finite-size effects are minimised. Determining the chemical potential in this way would have been computationally intractable had we employed ab initio calculations, and is only made possible by the use of the MLP. Once the chemical potential is known for the MLP at one set of conditions, we can find it at other pressures and temperatures by numerically integrating the Gibbs–Duhem relation along isotherms and the Gibbs-energy analogue of the Gibbs–Helmholtz relation along isobars, respectively.

To find the free energy of the liquid phase at the MLP level, we perform a series of direct-coexistence simulations[44,58] of the liquid in contact with ice $I_h$ at a series of temperatures at 0 bar to determine the coexistence temperature. The melting point $T_m = 279.5$ K at 1 bar for the MLP, although evaluated in a different way, is in perfect agreement with a previous simulation that used umbrella sampling[29]. We equate the chemical potentials of the two phases under these conditions and obtain the chemical potential of the liquid at other conditions by thermodynamic integration along isotherms and isobars, as for the ice phases.

In the second step of the procedure [Fig. 1(ii)], we promote the chemical potential of the system as described by the MLP to the DFT potential-energy surface level of theory using a free-energy perturbation,

$$\Delta\mu^{\text{MLP}\rightarrow\text{DFT}}(P,\ T) = -\frac{k_B T}{N}\ln\left\langle \exp\left[-\frac{U_{\text{DFT}} - U_{\text{ML}}}{k_B T}\right]\right\rangle_{P,T,\mathcal{H}_{\text{ML}}}, \quad (1)$$

where $\langle\cdot\rangle_{P,T,\mathcal{H}_{\text{ML}}}$ denotes the ensemble average of the system sampled at temperature $T$ and pressure $P$ using the ML hamiltonian $\mathcal{H}_{\text{ML}}$, where $U_{\text{DFT}}$ is the DFT energy and $U_{\text{ML}}$ is the MLP energy, and $N$ is the number of water molecules in the system. This correction is necessary to recover the true potential at the DFT level, in particular to compensate for the lack of long-range electrostatics, which are particularly important in modulating ice phase stabilities[59]. Using the MLP-derived chemical potentials without corrections neglects such long-range contributions and significantly hampers the quantitative accuracy of the computed phase diagram. In practice, we first collect trajectories of ice configurations by performing MD simulations in the isothermal-isobaric ensemble for each promising ice phase using the MLP. The number of water molecules in the simulation cell ranges from 56 to 96 molecules, depending on the unit-cell size of each phase. For each of the proton-disordered (I, VI) or partially disordered phases (III, V), we run independent simulations for 5–8 different depolarised proton-disordered configurations. We then select decorrelated configurations from the MD trajectories and recompute their energies at the DFT level. The MLP → DFT correction terms are then computed using Eq. (1) and averaged over the different disordered structures for the proton-disordered phases.

In this step, the correction to the chemical potential ranges from −7.9 meV (for ice VI at 125 K and 10,000 bar) to 10.8 meV (for ice XV at 275 K and 5000 bar). Ice XV is the proton-ordered analogue of ice VI; similarly, ice XIII, the proton-ordered analogue of ice V, is more stable at the MLP level than it ought to be [e.g. at 225 K and 7000 bar, the corrections to chemical potentials are $\Delta\mu_V = -1.5$ meV and $\Delta\mu_{\text{XIII}} = 3.4$ meV]. Indeed the difference in the correction term between these two pairs of phases is largely insensitive to temperature and pressure, at least at the pressures where the phases are competitive and for which we have computed the correction reliably; namely, $\Delta\Delta\mu(\text{V}\rightarrow\text{XIII}) \approx 5$ meV and $\Delta\Delta\mu(\text{VI}\rightarrow\text{XV}) \approx 11$ meV.

During the final TI [Fig. 1(iii)], NQEs are taken into account by integrating the quantum centroid virial kinetic energy $\langle E_k\rangle$ with respect to the fictitious 'atomic' mass $\bar{m}$ from the classical (i.e. infinite) mass to the physical masses $m$[47,60–63]. In practice, a change of variable $y = \sqrt{m/\bar{m}}$ is applied to reduce the discretisation

error in the evaluation of the integral[60], yielding

$$\Delta\mu^{\mathrm{NQE}}(P,\ T) = 2\int_0^1 \frac{\langle E_k(1/y^2)\rangle}{y}\ \mathrm{d}y. \qquad (2)$$

We evaluate the integrand using PIMD simulations for $y = 1/4$, $1/2$, $\sqrt{2}/2$ and 1, and then numerically integrate it. In these PIMD simulations, we use 24 beads for all phases considered at a wide range of constant pressure and temperature conditions for $T \geq 125$ K. Relative to ice II, the correction to the chemical potential arising from NQEs ranges from $-8$ meV to 5 meV.

**Uncertainty estimation**. To determine the approximate uncertainty associated with the individual chemical potential contributions, we first compute the reference chemical potential of several phases computed from an equilibrated perfect crystal at 125 K and 0 bar. We repeat this calculation between 5 and 10 times and determine the standard deviation of the resulting chemical potentials. For (partially and fully) proton-disordered phases, we split the contributions to the standard deviation arising from thermodynamic integration, which we obtain by computing the reference chemical potential calculation using the same initial configuration in 5 independent simulations, and from proton disorder, which we obtain by averaging over the chemical potentials arising from several distinct manifestations of the proton disorder in the initial configurations. Similarly, we compute the standard deviation of the correction to the chemical potential when the system is integrated from the MLP to the DFT level evaluated at 225 K and 3000 bar. For the correction to NQEs, we determine the statistical error in the measurement of the kinetic energy at each scaled mass. We calculate the integral of Eq. (2) with an analogue of Simpson's rule for irregularly spaced data both for the mean values as well as the mean value increased by the standard deviation, and we estimate the a posteriori NQE error to be the difference between these two corrections.

## Data availability

The 50 ice structures studied, along with their chemical potentials, the CP2K input files and a Mathematica notebook for analysis, are provided in Supplementary Data 1 and in the supporting data available at https://doi.org/10.17863/cam.62110 and https://github.com/BingqingCheng/nn-water-phase-diagram.

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

## Acknowledgements

We thank Jan Gerit Brandenburg and Carlos Vega for reading an early draft and providing constructive and useful comments and suggestions. A.R. and B.C. acknowledge resources provided by the Cambridge Tier-2 system operated by the University of Cambridge Research Computing Service funded by EPSRC Tier-2 capital grant EP/P020259/1. B.C. acknowledges funding from the Swiss National Science Foundation (Project P2ELP2-184408), and allocation of CPU hours by CSCS under Project ID s957.

## Author contributions

A.R. and B.C. designed and performed the research, analysed data and wrote the paper.

## Competing interests

The authors declare no competing interests.
