## [Peer Review File · Nature Communications]

REVIEWERS' COMMENTS

Reviewer #1 (Remarks to the Author):

This is a very interesting article on the theoretical prediction of the phase diagram of water based on a quantum mechanical description. The computation of the phase diagram of water is extremely challenging due to the variety of structure that can occur as well as the subtle effects of (partial) proton order and quantum fluctuations. In a technical tour de force, based on a combination of state of the art techniques including DFT, thermodynamic integration, machine learned potentials, thermodynamic perturbation theory and path integral molecular dynamics, the authors manage to compute the phase diagram of water over a wide pressure range. For three different DFT-models the resulting phase diagram is fairly close to the experimental one. I think this work is important for several (technical and non-technical) reasons:

- 1) It shows that DFT at the hybrid level is capable of describing water accurately over a wider temperature and pressure range.
- 2) Residual discrepancies between the computed and the experimental phase diagrams are most likely due to differences in the DFT functionals on which the calculations are based.
- 3) While the authors included over 50 candidate structure (i.e., many more than the experimentally known one), none of these phases were thermodynamically stable in the considered pressure-temperature range, indicating that most likely all stable phase have been found already (at least in the pressure and temperature range considered).
- 4) At high pressures nuclear quantum effects, which are often neglected, seem to play a major role in determining thermodynamics stability.

I think this paper is a major step forward and demonstrates what is possible with a smart combination of techniques. I support publication in Nature Communications.

I have a only a few things I'd like the authors to consider:

- In Fig. 2a there are many phases with a large offset on the y-axis, but with large slope (dashed and dotted lines). This indicates that at negative pressures these might become the preferred phases. Is this something the authors have considered or even investigated? Perhaps this is worth a comment in the paper.
- The lines corresponding to ice XI and XIV in 1a(ii) have disappeared in Fig 2b and 2c. Are they beyond the range shown or have they been omitted for other reasons?
- Similarly, phases VII and XI seem to be not shown in Fig. 2a. Or are they just not labelled?
- The hatched areas mentioned in the captions of Fig. 2 and Fig. 3 are actually hard to see in the figures.
- In Fig. 4 the color code is not clear. I cannot find the lighter shades of blue (ice V) and red (ice I) in the color bar on top of the figure. And why is liquid water shown in white? Also, according to the color bar the area of stability of ice II should be shown in pink.

Reviewer #2 (Remarks to the Author):

In this manuscript, the phase diagram of water is mapped out within density functional theory and compared to experiment. Specifically, three different hybrid density functionals are used to determine the most stable forms of ice for pressures between 0 to 10,000 bar and temperatures between 25 to 300 K. A previously described computational approach based on the use of a machine-learning potential and thermodynamic integration techniques is used to efficiently map out the ice phase diagram. Finally, the effects of nuclear quantum effects on the phase diagram of water are examined with the use of path integral sampling. The primary conclusion from this work is that the three density functionals investigated have limitations and the level of agreement with experiment is “very good, but by no means perfect”.

The sheer amount of computational work presented in this paper is impressive. The authors outline a detailed workflow that keeps the calculations trackable while also ensuring that the statistical sampling is sufficiently robust to describe the remarkably small energy differences between the candidate phases of ice. The computational approach outlined is quite general and can be used to map out the phase diagram of a broad range of materials and systems. Unfortunately, the paper is almost entirely focused on describing the overall computational approach (much of which has been presented before) and showing how the computed phase diagrams compare to experiment at a very superficial level. This is somewhat disappointing given that the abstract of this paper starts with the statement “The phase diagram of water harbors many mysteries: some of the phase boundaries are fuzzy, and the set of known stable phases may not be complete”. Despite this bold statement, the paper does not provide any new insight into these mysteries. The computational approach appears to be sensible and well thought out, but the reader is very much left with the impression that much more work needs to be done and the quality of the theoretical models (based on density functional theory) are simply not good enough to provide any new insight into the phase diagram of water.

After reading through the paper I am left with the impression that the work presented is an important stepping stone on the path to computing the phase diagram of water, but it is certainly not the full story. Unfortunately, I don't think that the main findings in this paper are significant enough to warrant publication in a journal like Nature.

We thank the referees for their careful reading of our manuscript and for their positive assessment. We have made a number of changes to the manuscript in response to their comments, and have highlighted these in blue in the revised version of the text. In what follows, we respond to each of the points raised by the referees.

Reviewer #1 (Remarks to the Author):

This is a very interesting article on the theoretical prediction of the phase diagram of water based on a quantum mechanical description. The computation of the phase diagram of water is extremely challenging due to the variety of structure that can occur as well as the subtle effects of (partial) proton order and quantum fluctuations. In a technical tour de force, based on a combination of state of the art techniques including DFT, thermodynamic integration, machine learned potentials, thermodynamic perturbation theory and path integral molecular dynamics, the authors manage to compute the phase diagram of water over a wide pressure range. For three different DFT-models the resulting phase diagram is fairly close to the experimental one. I think this work is important for several (technical and non-technical) reasons:

- 1) It shows that DFT at the hybrid level is capable of describing water accurately over a wider temperature and pressure range.
- 2) Residual discrepancies between the computed and the experimental phase diagrams are most likely due to differences in the DFT functionals on which the calculations are based.
- 3) While the authors included over 50 candidate structures (i.e., many more than the experimentally known one), none of these phases were thermodynamically stable in the considered pressure-temperature range, indicating that most likely all stable phases have been found already (at least in the pressure and temperature range considered).
- 4) At high pressures nuclear quantum effects, which are often neglected, seem to play a major role in determining thermodynamic stability.

I think this paper is a major step forward and demonstrates what is possible with a smart combination of techniques. I support publication in Nature Communications.

Authors:

We thank the referee for taking the time to review our work, and for their deep understanding of our approach.

I have a only a few things I'd like the authors to consider:

- In Fig. 2a there are many phases with a large offset on the y-axis, but with large slope (dashed and dotted lines). This indicates that at negative pressures these might become the preferred phases. Is this something the authors have considered or even investigated? Perhaps this is worth a comment in the paper.

Authors:

Indeed, the low-density ice phases may become favourable at negative pressures: in fact the dashed lines correspond to known low-density phases of ice, and so their large gradients in the chemical potential are expected. We have not attempted to characterise the phase behaviour at negative pressures; however, if we extrapolate the chemical potentials at 225 K to large negative pressures, we find that for the MLP, the sH clathrate hydrate structure becomes thermodynamically stable below approximately -5 kbar, while the sII clathrate hydrate structure takes over at approximately -5.6 kbar. However, these are fairly extreme extrapolations based on data obtained at positive pressures, so they are not likely to be very reliable. Moreover, we have not evaluated corrections to the DFT level of theory or NQEs for these phases. It would, however, be very interesting to investigate in future work the phase behaviour at negative pressures, since it is much less well explored and less accessible to experiment than the phase diagram at positive pressures. We have added a comment to the last paragraph of the Discussion section to clarify this; we have also added a reference to the literature where the phase diagram at negative pressures is discussed in more detail.

- The lines corresponding to ice XI and XIV in 1a(ii) have disappeared in Fig 2b and 2c. Are they beyond the range shown or have they been omitted for other reasons?

Authors:

Both ices XI and XIV are proton-ordered phases, and they were judged as uncompetitive in our study at 225 K since (i) the MLP over-stabilises the proton-ordered phases relative to DFT (typically by about 10 meV under these conditions), and (ii) ice I (the proton-disordered analogue of ice XI) is more stable than ice XI already at the MLP level. We have now clarified why these phases are not shown when DFT corrections are applied in the manuscript (in paragraph 4 of the 'Chemical potentials' section).

- Similarly, phases VII and XI seem to be not shown in Fig. 2a. Or are they just not labelled?

Authors:

The phase XI was shown in Fig. 2a in yellow, which may not have been visible enough. We have changed this colour to light blue to make it stand out better.

Ice VII is in fact also shown in Fig. 2a(i): it is the light green line starting at approximately 50 meV and ending at approximately 30 meV. Like all the other lines in Fig. 2a(i), it is not labelled, and it does not appear in Fig. 2a(ii) since its chemical potential is outside the range of the figure. However, we now briefly mention ice VII in the last paragraph of the Discussion section, since its phase transition with liquid water is particularly interesting, as it has not been possible to capture with any empirical pair potential.

- The hatched areas mentioned in the captions of Fig. 2 and Fig. 3 are actually hard to see in the figures.

Authors:

In Fig. 2, we have improved the contrast between the 'solid' area and the colour of the hatchings, and we have moreover made the hatched area lighter than the remaining bars, so that the hatching is no longer the only distinguishing feature. We have made a note of this in the caption.

To make the hatched area stand out more in Fig. 3, we have changed the colour associated with ice III and the colour of the II/V coexistence line (across all figures for consistency). The hatched region in Fig. 3 now contrasts better with the neighbouring colours. We have changed the order in which the hatchings are drawn: they are now plotted last, i.e. on top of the coexistence lines.

- In Fig. 4 the color code is not clear. I cannot find the lighter shades of blue (ice V) and red (ice I) in the color bar on top of the figure. And why is liquid water shown in white? Also, according to the color bar the area of stability of ice II should be shown in pink.

Authors:

We were a bit mystified by this comment until we opened the PDF in a few different viewers, and then we understood what the referee meant. Unfortunately it appears that the colour bar was not rendered correctly in certain PDF viewers (such as the internal PDF viewer in Firefox), with white replaced by purple for reasons unknown, even though it was rendered as intended in viewers such as Okular and Adobe Reader. We have now produced the colour bar using a different method, and for us it now opens correctly in all the PDF viewers we have tried. We will ensure that at the proof stage, we open the publisher-produced PDF in a wide range of viewers to make sure that this does not happen again.

[Further to our response to the previous point, when investigating the issue with the colour bar, we have also found that there is a bug in the internal Firefox pdf.js viewer [issue #11473], which thus renders hatchings incorrectly; this does unfortunately make the hatching seem less prominent. Perhaps the referee happened to use this PDF viewer? Although this is an issue with the PDF viewer rather than the figures themselves, we have made other changes, as detailed above, which make the hatchings less important, and we believe the figures are now sufficiently clear even in the Firefox PDF viewer.]

Reviewer #2 (Remarks to the Author):

In this manuscript, the phase diagram of water is mapped out within density functional theory and compared to experiment. Specifically, three different hybrid density functionals are used to determine the most stable forms of ice for pressures between 0 to 10,000 bar and temperatures between 25 to 300 K. A previously described computational approach based on the use of a

machine-learning potential and thermodynamic integration techniques is used to efficiently map out the ice phase diagram. Finally, the effects of nuclear quantum effects on the phase diagram of water are examined with the use of path integral sampling. The primary conclusions from this work is that the three density functionals investigated have limitations and the level of agreement with experiment is “very good, but by no means perfect”.

AUTHORS:

We thank the referee for taking the time to review our work and for their insightful reading of our work.

The sheer amount of computational work presented in this paper is impressive. The authors outline a detailed workflow that keeps the calculations trackable while also ensuring that the statistical sampling is sufficiently robust to describe the remarkably small energy differences between the candidate phases of ice. The computational approach outlined is quite general and can be used to map out the phase diagram of a broad range of materials and systems. Unfortunately, the paper is almost entirely focused on describing the overall computational approach (much of which has been presented before) and showing how the computed phase diagrams compare to experiment at a very superficial level. This is somewhat disappointing given that the abstract of this paper starts with the statement “The phase diagram of water harbors many mysteries: some of the phase boundaries are fuzzy, and the set of known stable phases may not be complete”. Despite this bold statement, the paper does not provide any new insight into these mysteries. The computational approach appears to be sensible and well thought out, but the reader is very much left with the impression that much more work needs to be done and the quality of the theoretical models (based on density functional theory) are simply not good enough to provide any new insight into the phase diagram of water.

Authors:

We have removed the phrase about the ‘mysteries’ of the phase diagram of water, so that the abstract is now more clearly outlining the computational approach developed in the main text.

After reading through the paper I am left with the impression that the work presented is an important stepping stone on the path to computing the phase diagram of water, but it is certainly not the full story. Unfortunately, I don’t think that the main findings in this paper are significant enough to warrant publication in a journal like Nature.

Authors:

We agree that there is a lot of future work to do, both for the water phase diagram and for studying the phase behaviour of other systems, and our work serves as an important stepping stone towards fully addressing these important problems.